# Estimating the time-varying effective reproduction number via Cycle Threshold-based Transformer

Xin-Yu Zhang[1,2], Lan-Lan Yu[1,2], Wei-Yi Wang[1,2], Gui-Quan Sun[3,4]*, Jian-Cheng Lv[1,2], Tao Zhou[5], Quan-Hui Liu[1,2]*

**1** College of Computer Science, Sichuan University, Chengdu, China, **2** Engineering Research Center of Machine Learning and Industry Intelligence, Ministry of Education, Sichuan University, Chengdu, China, **3** Department of Mathematics, North University of China, Taiyuan, China, **4** Complex Systems Research Center, Shanxi University, Taiyuan, China, **5** Big Data Research Center, University of Electronic Science and Technology of China, Chengdu, China

* gquansun@126.com (G-QS); quanhuiliu@scu.edu.cn (Q-HL)

**Data Availability Statement:** Data and code are available on Github (https://github.com/xiaohuahua1/Ct-Transformer).

## Abstract

Monitoring the spread of infectious disease is essential to design and adjust the interventions timely for the prevention of the epidemic outbreak and safeguarding the public health. The governments have generally adopted the incidence-based statistical method to estimate the time-varying effective reproduction number $R_t$ and evaluate the transmission ability of epidemics. However, this method exhibits biases arising from the reported incidence data and assumes the generation interval distribution which is not available at the early stage of epidemic. Recent studies showed that the viral loads characterized by cycle threshold (Ct) of the infected populations evolving throughout the course of epidemic and providing a possibility to infer the epidemic trajectory. In this work, we propose the Cycle Threshold-based Transformer (Ct-Transformer) to estimate $R_t$. We find the supervised learning of Ct-Transformer outperforms the traditional incidence-based statistic and Ct-based $R_t$ estimating methods, and more importantly Ct-Transformer is robust to the detection resources. Further, we apply the proposed model to self-supervised pre-training tasks and obtain excellent fine-tuned performance, which attains comparable performance with the supervised Ct-Transformer, verified by both the synthetic and real-world datasets. We demonstrate that the Ct-based deep learning method can improve the real-time estimates of $R_t$, enabling more easily adapted to the track of the newly emerged epidemic.

## Author summary

The time-varying effective reproduction number $R_t$ is an important indicator in tracking the epidemic spread. The well-known method to estimate $R_t$ is the incidence-based statistical method, which is constrained with the assumptions and available data. The recent studies show that the time-varying distribution of cycle threshold (Ct) values of the sampled infected population provides a possibility to infer the epidemic trajectory. Here, we propose the Cycle Threshold-based Transformer (Ct-Transformer), a deep neural

**Funding:** Q. H. L recevies the funding from the National Natural Science Foundation of China (url: https://www.nsfc.gov.cn/) with grant number 62373264, and the Major Program of National Fund of Philosophy and Social Science of China (url: http://www.nopss.gov.cn/GB/index.html) with grant number 20&ZD112. The funders had no role in study design, data collection and analysis, decision to publish, or preparation of the manuscript.

**Competing interests:** The authors have declared that no competing interests exist.

network based method to estimate $R_t$. The results on both the synthetic and real-world datasets demonstrate that the Ct-Transformer surpass the traditional incidence-based and the existing Ct-based estimating methods. More importantly, the proposed self-supervised learning of Ct-Transformer estimates the $R_t$ accurately for the newly emerged infectious disease. Our study suggests that Ct-based deep learning method can be employed to improve the tracking of the spread of infectious disease, and especially for the newly emerged epidemic.

## Introduction

The emergence of epidemics, including the Corona Virus Disease 2019 (COVID-19) pandemic [1], presents a serious risk to public health and even human lives. Real-time tracking of an emerging infectious disease is essential to inform better control policies and avoid a large number of infections. The government generally have used the estimates of the time-varying effective reproduction number $R_t$ to monitor the transmission of an epidemic, as it provides insights into the transmission risk and the assessment of implemented interventions timely [1–6]. In specific, the time-varying effective reproduction number $R_t$ can be defined in two ways: the instantaneous reproductive number and the case reproductive number [7]. Therein, the case reproduction number is defined as the average number of secondary cases generated by an infectious individual infected at time $t$ [8], which is used in this work. $R_t < 1$ suggests that the epidemic is waning and can be considered that the transmission of epidemic is under control [9].

Most existing studies [1, 10–17] have adopted the statistical methods to estimate $R_t$ based on the time series of daily symptomatic cases, hospitalizations and death numbers. Wallinga and Teunis [10] propose a likelihood-based method (WT method) to estimate $R_t$ based on the daily number of reported cases and the generation interval distribution. The estimation results of this method may show substantial variation in a short period when the data aggregation time step is small. As an improvement, Cori et al. [13] develop the EpiEstim, a generalized R toolkit that estimates $R_t$ based on Bayesian inference. The EpiEstim provides precise posterior distribution of $R_t$ and has been employed to monitor several recent outbreaks [17–20]. Gressani et al. [14] propose the EpiLPS, which can smooth the epidemic curve and allows to obtain accurate estimates of $R_t$. Liu et al. [15] propose a discrete spline-based approach rtestim and produce a locally adaptive estimation results for $R_t$. Parag et al. [16] construct a recursive inference algorithm and develop the EpiFilter. This method unifies the WT method and EpiEstim and largely resolves their edge-effect issues. These methods typically require the assumption of the generation interval distribution (i.e., the time interval between the infection time of the infector and her/his infectees), which is hard to acquire at the early stage of the emergent epidemic [7, 21]. Further, the aforementioned time series data is sensitive to detection resources, latent periods, and reporting delays [22–24].

Recent studies [25, 26] demonstrate that the viral load distribution of infected populations can be used to infer the trajectory of epidemics. They show that the fast-growing epidemiological populations exhibit a predominance of new infections who have higher viral loads, while the shrinking epidemiological populations have more older infections who have lower viral loads. The viral load of an infected individual can be represented by the cycle threshold (Ct) values, which are obtained from the real-time quantitative reverse-transcription polymerase chain reaction (RT-qPCR) assay and the lower Ct value indicates a higher viral load [27, 28]. Some works [25, 29–33] have attempted to estimate $R_t$ using the statistical methods by

monitoring the time-varying distribution of Ct values of the infected populations. Hay et al. [25] use the Ct values from cross-sectional samples to estimate $R_t$ based on Bayesian inference. They also develop the R toolkit ViroSolver to infer epidemic dynamics, including the estimates of $R_t$. However, this method makes specific assumptions, such as the epidemic trajectory following a Gaussian process. In a follow-up study, Hay et al. [29] explore the epidemic dynamics of COVID-19 variants of concern (VOCs) and expand the capabilities of the ViroSolver to estimate $R_t$ in a two-strain epidemic. Liu et al. [34] compare the EpiEstim [13] and ViroSolver [25] and conclude that the ViroSolver provides more accurate estimates of $R_t$, while the EpiEstim requires the adjustment of the generation interval distribution for better performance. Besides, Lin et al. [31] apply a log-linear regression model to estimate $R_t$ based on the mean and skewness of Ct values in the infected populations. They show that the Ct-based $R_t$ estimation methods are less sensitive to the detection resources [28]. The above Ct-based $R_t$ estimation method adopts statistical techniques, such as Bayesian and regression models. Although effective, these methods are constrained by their reliance on assumptions and limited capability to extract the temporal features from the time-varying distribution of Ct values and process the non-linear relationships.

With the increasing availability of data and powerful learning ability, the deep learning as a nonlinear mathematical method is widely used in image recognitions [35], natural language processing [36], and time series forecasting [37], etc. The core of deep learning is to build machine learning architecture model with multiple hidden layers, to train them on large-scale data, and obtain a large amount of more representative feature information [38]. And then, some samples can be used to test the model and finally improve the model performance such as the prediction accuracy. The early notable deep learning models include Recurrent Neural Networks (RNNs) [39] and Long Short-Term Memory (LSTM) networks [39]. Therein, the architecture of RNNs is designed to extract temporal feature from sequence data, while the LSTM, as an advanced form of RNNs, is built for capturing long-term dependencies in sequence data. During the COVID-19 period, the deep learning models have been applied in the field of epidemiology [40–43]. Davahli et al. [40] develop two types of Graph Neural Networks (GNN) models [44], which are designed to handle graph-structured data, to estimate $R_t$ using COVID-19 cases from 22 January 2020 to 26 November 2020 in the United States. Gatto et al. [41] utilize LSTM architecture to predict the daily trend of time-varying reproduction number ($R_t$). They find the neural network-based models accurately predict $R_t$ in the region and autonomous province of Italy when sufficient epidemiological data is available. Although the above architectures of neural network are effective in time series prediction tasks, they still exhibit the vanishing gradients and hard to capture long-range dependencies. A significant advancement in modern machine learning is that Vaswani et al. [45] propose the Transformer, an encoder-decoder based neural network architecture with the novel self-attention mechanism. The encoder is designed to process the input sequence, and the decoder generates output sequences, and both of them consist multiple layers and communicating via attention mechanism. The self-attention mechanism allows the model to weigh the importance of different parts of the input sequence and enables it to capture the long-range dependencies efficiently. Both the computation efficiency and its ability to handle with long-range dependencies have made the Transformer architecture a mainstream choice in deep learning models.

The main aim of this study is to estimate the case reproduction number from the time-varying distribution of Ct values based on deep learning models. In order to better capture the long-range dependencies and extract the temporal features, we propose the Cycle Threshold-based Transformer (i.e., Ct-Transformer) with the canonical components, including the patching layer, the gated recurrent unit network and the multi-head attention layer. The detailed descriptions of the above components in the architecture of Ct-Transformer are presented in

Materials and methods. To be noted that, both the supervised learning and the self-supervised learning of Ct-Transformer are developed. Therein, the supervised learning of Ct-Transformer is trained on a large number of datasets with labels, while the self-supervised one can be trained on the datasets without labels and fine-tuning the pre-trained model on small portion of labeled data, which finally both of them can be used to estimate $R_t$. By evaluating the Ct-Transformer on the synthetic datasets, we find the supervised learning of Ct-Transformer performs best in all Ct-based deep learning and statistical methods. More in detail, the supervised learning of Ct-Transformer achieves a 31.1% reduction in MAE and a 27.0% reduction in RMSE, compared with other cycle thresholds based deep learning methods. We also find the proposed Ct-Transformer is robust to both the limited and the time-varying detection resources. It achieves similar performance or better performance in the situations with 25% detection rate compared with the traditional incidence-based $R_t$ estimation method in the situations with 100% detection rate. Lastly, we pre-train the self-supervised learning of Ct-Transformer on synthetic datasets, and then fine-tune the pre-trained model on the Hong-Kong COVID-19 data. We obtain a 24% reduction in MAE compared with the result reported in [31].

In this study, we propose the Cycle Threshold-based Transformer (Ct-Transformer), which is a deep learning-based method to estimate $R_t$. The self-supervised learning of Ct-Transformer can be used to estimate $R_t$ of the same disease on other regions or countries by fine-tuning on small portion of real labeled data. The proposed Ct-Transformer is evaluated on both the synthetic and real datasets, which proves to be effective and robust to the detection resources. The ablation experiments also show the necessity of each canonical components in the design on the architecture of Ct-Transformer. In summary, this study offers public health authorities a deep learning method for estimating $R_t$ without the time delays inherent in traditional incidence-based methods and few prior knowledge requirement. Additionally, it can be easily adapted to estimate $R_t$ for newly emerging diseases by adjusting the model with small portion of real data.

## Materials and methods

### The supervised learning of Ct-Transformer

We consider the following problem: given a time series of Ct variables with length $L$: $(\vec{x}_1, \vec{x}_2, \ldots, \vec{x}_L)$, where each $\vec{x}_t$ is a vector with dimension $M$ (i.e., $\vec{x}_t = [x_t^1, x_t^2, \ldots, x_t^M]$, $1 \leq t \leq L$), we estimate the corresponding time series of $R_t$ with length $L$: $(R_1, R_2, \ldots, R_L)$. The dimension $M$ refers to the number of Ct variables, including the mean, skewness and distribution of Ct values. Detailed exploration of dimension $M$ is shown in S1 Supplementary Methods (3. Ct Variables in the Synthetic Datasets). The workflow of Ct-Transformer and an overview of the architecture of supervised learning of Ct-Transformer are respectively as shown in Fig 1A and 1B.

**Patching layer.** The Patching layer segments the input $(\vec{x}_1, \vec{x}_2, \ldots, \vec{x}_L)$ into $N$ non-overlapping patches, and outputs a time series of patch $(\vec{s}_1, \vec{s}_2, \ldots, \vec{s}_L)$, where $N$ is the number of patches, calculated as $N = \lceil \frac{L}{P} \rceil$. $P$ represents the length of each patch and the dimension of the $i$−th patch $\vec{s}_i$ is $C = M{\times}P$, with $1 \leq i \leq N$. To be noted that, we append $P - (L \bmod P)$ repeated instances of the last value $\vec{x}_L \in \mathbb{R}^M$ to the end of the time series of Ct variables before patching. The patching layer shortens the length of the input from $L$ to $L/P$, which makes the Ct-Transformer able to deal with long time series and also improves computational efficiency. The "Prediction Head" at the end of Ct-Transformer is designed to flatten the series back to its original length.

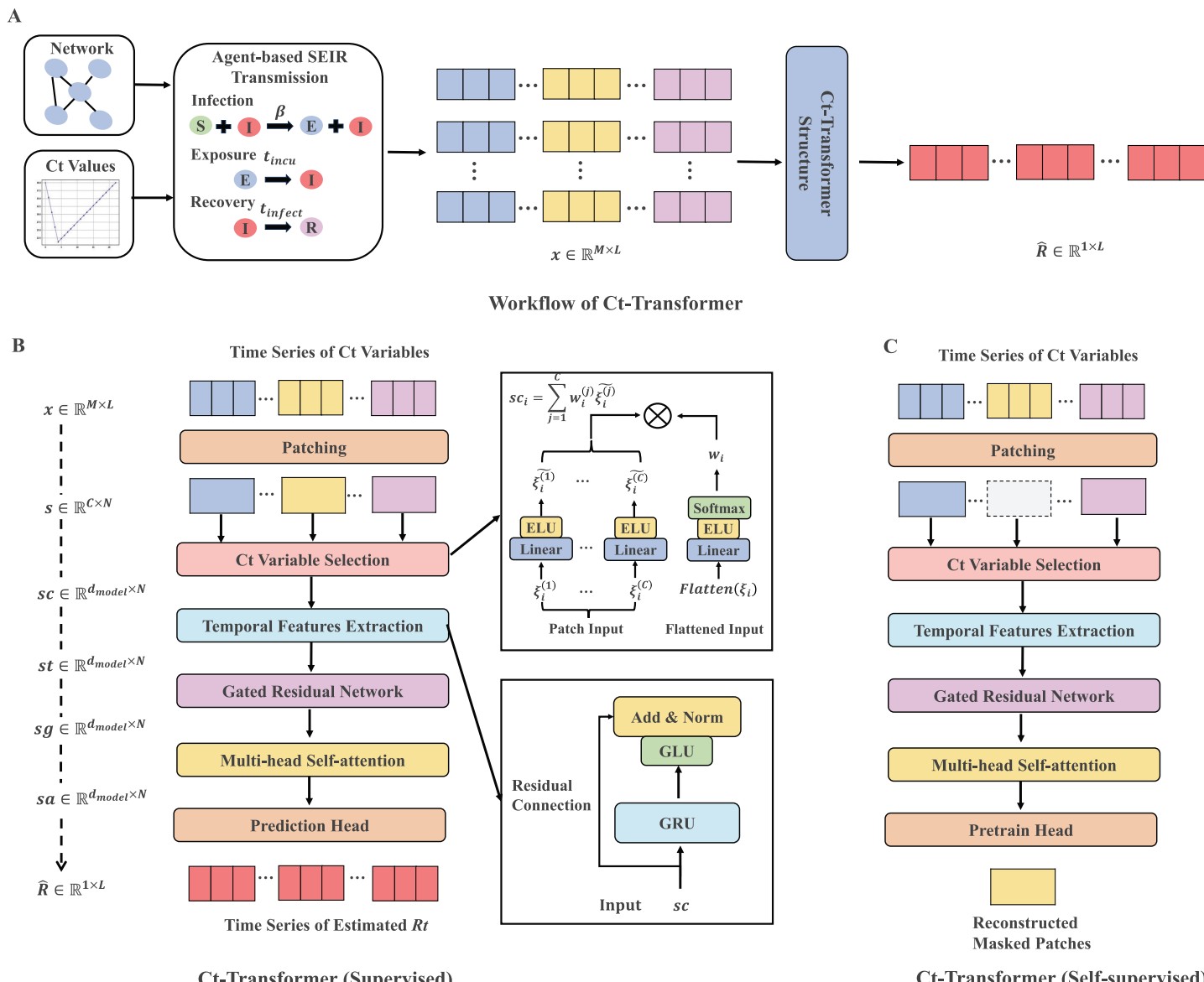

**Fig 1. The architecture of Ct-Transformer.** (A) The Ct-Transformer accepts a time series of Ct variables and outputs the corresponding time series of estimated $R_t$. The agent-based SEIR transmission method is designed for producing the synthetic datasets. (B) The Ct-Transformer segments the time series of Ct variables into patches, which are then projected into a high-dimensional space based on their respective weights. The model extracts the temporal features from the high-dimensional hidden representations and builds non-linear relationships with $R_t$. (C) Masked self-supervised learning of the Ct-Transformer, where patches are randomly selected and set to zero. The Ct-Transformer is then tasked with reconstructing these masked patches.

**Ct variable selection layer.**  Inspired by the Variable Selection Network [46] and the Temporal Covariate Interpreter [47] designed in the neural network models, we introduce the Ct Variable Selection (CVS) layer in the architecture of Ct-Transformer to identify the important variables by calculating the weight of variables. The CVS layer first transforms each variable of the output from the patching layer into a high-dimensional space. Let $\xi_i^{(j)} \in \mathbb{R}^{d_{model}}$ denote the representation of the $j$-th variable in the $i$−th patch, where $d_{model}$ represents the dimension of representation. The representations of all variables in the $i$−th patch are flattened and

expressed as $Flatten(\xi_i) = [\xi_i^{(1)^T}, \xi_i^{(2)^T}, \ldots, \xi_i^{(C)^T}]^T$. Further, a linear transformation layer is employed to obtain the weights of all variables in the $i$–th patch, which is listed as:

$$w_i = Softmax(ELU(W_1 * Flatten(\xi_i) + b_1)) \tag{1}$$

where $w_i \in \mathbb{R}^C$ represents the weights of all variables in the $i-$th patch. ELU and $Softmax(\cdot)$ respectively represent the Exponential Linear Unit activation function [48] and the Softmax operation [49]. $W_1$ and $b_1$ respectively represent the learnable weights and biases. Finally, each representation $\xi_i^{(j)}$ is operated by a linear transformation and then weighted by its weight. Thus, the output of the CVS layer is expressed as follows:

$$sc_i = \sum_{j=1}^{C} w_i^{(j)} * ELU(W_2 * \xi_i^{(j)} + b_2) \tag{2}$$

where $W_2$ and $b_2$ respectively represent the learnable weights and biases.

**Temporal Features Extraction layer.** The Temporal Features Extraction (TFE) layer designed in Ct-Transformer incorporates a gated recurrent unit (GRU) network [50] to extract the temporal features from the output of the CVS layer. The GRU network can process and remember information from long time series data more effectively than the traditional RNNs [39]. This network employs two critical gates, i.e., reset gate and update gate, which regulate the information flow of the hidden state. The hidden states of all time steps from the GRU are passed to the latter residual connection. The residual connection integrated with a Gated Linear Unit (GLU) [51] further mitigates the problem of vanishing gradient and bolsters the process of training.

**Gated Residual Network Layer.** The Gated Residual Network (GRN) layer, implemented in the architecture of the Ct-Transformer, is mainly used for non-linear processing. This layer first applies two linear transformation layers with ELU activation to the output from the TFE layer, which results in $\tilde{sg} \in \mathbb{R}^{N \times d_{model}}$. A Gated Linear Unit is then designed to control the flow of information, which is expressed as follows:

$$GLU(\tilde{sg}) = \sigma(W_3 * \tilde{sg} + b_3) \odot (W_4 * \tilde{sg} + b_4) \tag{3}$$

where $W_3, W_4$ and $b_3, b_4$ respectively represent the learnable weights and biases of the linear layers. $\sigma(\cdot)$ and $\odot$ respectively represent the Sigmoid activation function [52] and the Hadamard product of the elements [51]. To further enhance the model's generalization, LayerNorm [53] is applied and produces the final output of the GRN layer (i.e., $sg \in \mathbb{R}^{N \times d_{model}}$).

**Multi-head Self-attention layer.** We develop the Multi-head Self-attention (MSA) layer to learn dependencies between time steps based on the self-attention mechanisms [45]. Inspired by the modification of the self-attention mechanisms in the TFT, the MSA layer adopts the shared $V$ matrix across each head and employs an additive aggregation of all heads, which is calculated as:

$$MultiHeadAttention(Q, K, V) = \tilde{H} W_H, \tag{4}$$

where

$$\tilde{H} = \{\frac{1}{h_m} \sum_{h=1}^{h_m} A(Q W_Q^{(h)}, K W_K^{(h)})\} V W_V \tag{5}$$

therein, $Q = K = V \in \mathbb{R}^{N \times d_{attn}}$ respectively represent the Query, Key and Value matrices and $d_{attn}$ is calculated as $d_{model}/h_m$. $h_m$ represents the number of heads and $W_H \in \mathbb{R}^{d_{attn} \times d_{model}}$ is used

for linear transformation to the output. $A(\cdot)$ represents the scaled dot-product attention [45], calculated as $A(Q, K) = Softmax\left(\frac{QK^T}{\sqrt{d_{attn}}}\right)$. $W_K^{(h)} \in \mathbb{R}^{d_{model} \times d_{attn}}$, $W_Q^{(h)} \in \mathbb{R}^{d_{model} \times d_{attn}}$ are the head-specific weights for the Keys and Queries, while $W_V \in \mathbb{R}^{d_{model} \times d_{attn}}$ are the head-shared weights for the Value.

From Eq (5), each head is capable of learning different attention patterns $A(QW_Q^{(h)}, KW_K^{(h)})$, which concentrate on the shared $V$ matrix. This process can be explained as an ensemble of attention patterns and enhances the representational capacity. Besides the Multi-Head Attention, a feedforward network with two linear layers is employed. Both the Multi-Head Attention and the feedforward network are followed by a residual connection with a GLU.

**Loss function.** The Ct-Transformer is optimized by minimizing the sum of the quantile losses $L_q$ [54], with respect to multiple quantile points $Q$ for all time step $T$:

$$L_{total} = \sum_{t \in T}\sum_{q \in Q} L_q(y_t, \hat{y}_t^q),$$
(6)

where

$$L_q(y_t, \hat{y}_t^q) = max(q(y_t - \hat{y}_t^q), 0) + max((1 - q)(\hat{y}_t^q - y_t), 0)$$
(7)

therein, $y_t$ and $\hat{y}_t$ respectively represent the ground truth of $R_t$ and estimated $R_t$ by the model. $Q$ represents the set of quantile points and is set as $Q$ = {0.025, 0.5, 0.975} to obtain the 95% confidence intervals in this paper.

## The self-supervised learning of Ct-Transformer

The training of supervised learning method requires a sufficient dataset with labels, which is not available for the emergence of new infectious diseases. As an improvement, we also propose the self-supervised learning of Ct-Transformer, which requires less labeled data for representation learning. This approach is particularly suited for the newly emergent epidemic or the spread of the same infectious disease in different countries with distinct contact patterns. As shown in Fig 1C, both kinds of learning of Ct-Transformer share the same architecture but with different training processes. Specifically, the "Prediction Head" in the supervised learning of Ct-Transformer is replaced by the "Pretrain Head" in the self-supervised learning.

The training of self-supervised learning includes two steps: (1) Masked self-supervised training to obtain a pre-trained model using a relatively large dataset without labels. During this initial step, portions of the input are randomly removed, and the model is trained with quantile loss to reconstruct these omitted parts [55]. This step is crucial for learning high-level abstract representations. (2) Supervised fine-tuning this pre-trained model on part of the target dataset with labels. Upon the completion of these two steps, the fine-tuned model is capable of estimating $R_t$ in the target dataset. Usually, there are two strategies for fine-tuning [56]: (a) linear probing (Lin. Prob) and (b) end-to-end fine-tuning (End2End). For (a), we only train the "Prediction Head", while freezing the rest of the network. For (b), we first update the "Prediction Head" using the Lin. Prob and then perform fine-tuning to the entire network.

## SEIR transmission model and the Ct value model

We adopt the agent-based stochastic Susceptible-Exposed-Infectious-Removed (SEIR) transmission model [57] and the Ct value model [25] to produce the synthetic dataset. The SEIR transmission model classifies the population into four types: susceptible ($S$), exposed ($E$), infectious ($I$) and removed ($R$). Therein, the susceptible state represents an individual who is not

infected but can be infected, the exposed state represents an individual who is infected but not infectious, and the infectious state represents an individual who is infected and can infect susceptible individuals. Lastly, the removed state represents an individual who has recovered from the infectious state and can not be infected again. The flow of these states is presented in Fig 1A.

We employ the Ct value model reported in [25] to simulate the evolution of viral load (characterized by the Ct value) within host since the infection. In specific, we denote the observed Ct value of an infected individual $a$ days after his/her infection as $Ct(a)$, and it is described in S1 Supplementary Methods (1. Parameters of the Agent-based SEIR Transmission Model and Ct Value Model). As shown in S1 Fig, the trajectories of Ct values capture the variation resulting from both swabbing variability and individual-level differences in viral kinetics.

The parameters used to simulate both the SEIR model and the Ct value model are listed in S1 Table in S1 Supplementary Methods (1. Parameters of the Agent-based SEIR Transmission Model and Ct Value Model).

## Data

**Synthetic datasets.** The synthetic dataset are generated with the same framework but with different transmission parameters. For each simulation, the data is produced with following three steps: 1) On each day of the simulation, we simulate the SEIR transmission model with one time step (i.e., one day) and produce the newly infected individuals at day $t$; 2) A subset of individuals are sampled on each day of the simulation determined by the detection scenarios described at the subsection of Impact of Detection Rates on Performance in 3. 3) For each individual $i$ sampled on day $a$, a Ct value $Ct(a - t)$ is generated according to the Ct value model. Therein, $a - t$ represents the elapsed time since the infection of individual $i$, which is described in S1 Supplementary Methods (1. Parameters of the Agent-based SEIR Transmission Model and Ct Value Model). To be noted that, at step 1), the micro-transmission chains including the time of each infected individual, and his/her infectees, which are used to calculate the time-varying effective reproduction number $R_t$ are recorded. The details about the calculation are presented in S1 Supplementary Methods (2. Calculation of $R_t$ based on Micro-transmission Chains).

We consider multiple scenarios by modeling outbreaks on two distinct types of contact networks: the Erdős-Rényi (ER) network [58] and Scale-Free (SF) network [59]. The ER network is a random graph where edges between nodes are formed independently and randomly, and is used as a theoretical baseline to study the spread of disease on networks. The degree of a node, i.e., the number of neighbors or edges connected to a node in the ER network, approaches a Poission distribution, which is relatively narrow and most of nodes in the ER network having similar degrees. Thus, the ER network is used to characterize the homogenous contact pattern of population. In contrast, the SF network is developed to characterize the degree distribution of real networks, which usually follows the power law. It means that a small number of nodes in the SF network have a very high number of connections, while most other nodes have relatively few connections. We set the average degree of both type of contact network equal to 10 and simulate the outbreaks with different value of $R_0$ (the average number of secondary cases generated by a typical infectious individual over the entire course of the infectious period in a fully susceptible population) to represent the modeled epidemic with different transmission abilities. As shown in Table 1, the synthetic datasets are classified into the ER and SF datasets. The Ct-Transformer is trained and evaluated on the training and validation sets in the ER (SF) dataset with $R_0$ equal to 1.5, 2.0, 2.5, and 3.0. The performance of the model is then tested on the separate testing set in the ER (SF) dataset, where $R_0$ equal to 1.2,

**Table 1. Detailed partitioning of the synthetic datasets, including the ER dataset and SF dataset.**

| Dataset Type | Split Dataset | Contact Network | $R_0$ Set | Simulations per $R_0$ | Total |
|---|---|---|---|---|---|
| ER dataset | Training set | ER | {1.5, 2.0, 2.5, 3.0} | 1200 | 4800 |
|  | Validation set |  |  | 240 | 960 |
|  | Testing set |  | {1.2, 1.8, 2.2, 2.8, 3.4} | 96 | 480 |
| SF dataset | Training set | SF | {1.5, 2.0, 2.5, 3.0} | 1200 | 4800 |
|  | Validation set |  |  | 240 | 960 |
|  | Testing set |  | {1.2, 1.8, 2.2, 2.8, 3.4} | 96 | 480 |

$R_0$ **Set:** a set containing multiple $R_0$; **Simulations per** $R_0$: number of simulations for each $R_0$; **Total:** total number of simulations.

1.8, 2.2, 2.8, and 3.4. The total number of simulations used to train, evaluate and test for the Ct-Transformer are 4800, 960, and 480, respectively. Detailed combinations of Ct variables are shown in S1 Supplementary Methods (3. Ct Variables in the Synthetic Datasets).

**Real-world dataset.** In addition to the synthetic datasets, we also evaluate the performance of the Ct-Transformer on the Real-world dataset (i.e., Hong Kong COVID-19 dataset). The Hong Kong COVID-19 dataset includes two consecutive waves of the spread of the COVID-19 in Hong Kong, which are respectively as the third wave spanning July to August 2020 and the fourth wave from November 2020 to March 2021. During these two periods, 8646 COVID-19 cases are detected through both the clinical diagnosis and the public health surveillance. In the meanwhile, the first available records of Ct values (derived from RT-qPCR tests targeting E gene) for the detected COVID-19 cases are collected. We align the dates of first available record of Ct value of all detected cases and compute the mean and the skewness of Ct values at each date during the third and forth waves of the spread of COVID-19 in Hong Kong. Besides, the estimated time series of $R_t$ reported in [31] are included in this Real-world dataset. The section of Real-world Data Verification provides a comprehensive description of the temporal changes in the mean and skewness of Ct values, as well as the associated $R_t$. To test the performance of Ct-Transformer on this real-world dataset, we adjust the input format of the Ct-Transformer with the mean and the skewness of Ct values at each time step. We adopt the same method reported in [31] and divide the real-world dataset into the training, validation and testing sets, which are respectively as follows:

1. Training set, from Jul 6th 2020 to Sept 17th 2020.

2. Validation set, from Sept 18th 2020 to Nov 15th 2020.

3. Testing set, from Nov 16th 2020 to Mar 23rd 2021.

## Experimental settings

**Data preprocessing.** We utilize the Mix-Max normalization technique, as detailed in [60], to standardize the non-uniform input data. This process ensures that no single variable disproportionately influences the model's performance. For instance, the average of Ct values for infected individuals is between 16 and 40, whereas their probability distributions span from zero to one. To address this, each input variable is normalized according to the following procedure:

$$x'_k = \frac{x_k - x_k^{min}}{x_k^{max} - x_k^{min}} \qquad (8)$$

**Table 2. Hyperparameters, corresponding tuning spaces, and the best hyper-parameter settings for the Ct-Transformer on the ER dataset and SF dataset.**

| Hyperparameter | Tuning Space | Best for ER dataset | Best for SF dataset |
|---|---|---|---|
| $d_{model}$ | 256,512 | 512 | 512 |
| $h_m$ | 2,4,6,8,10 | 4 | 4 |
| $N_{attn}$ | 1,2,3,4,5,6 | 3 | 2 |
| $d_{ff}$ | 1024,2048,4096 | 2048 | 1024 |
| $p_{drop}$ | 0.1,0.2,0.3,0.4 | 0.1 | 0.1 |
| $lr$ | 5e-4,5e-5,5e-6 | 5e-4 | 5e-5 |
| $N_{batch}$ | 16,32,64,128 | 16 | 16 |

$d_{model}$: hidden size in the CVS, TFE, GRN, and MSA; $h_m$: number of heads in the Multi-head Attention; $N_{attn}$: number of layers in the MSA; $d_{ff}$: dimension of the second linear layer in the feedforward network of the MSA; $p_{drop}$: dropout rate; $lr$: learning rate; $N_{batch}$: batch size.

where $x_k^{min}$ and $x_k^{max}$ respectively represent the minimum and maximum values of the $k$−th variable for the input in the training set. Thus, Min-Max Normalization transforms a value $x_k$ of the $k$−th variable to $x_k'$ in the range of zero to one. Besides, we apply the logarithmic transformation [61] to the value of $R_t$ to make the distribution of the label smoother and improve the training efficiency of Ct-Transformer.

**Evaluation metrics.** Mean absolute error (MAE), root mean square error (RMSE), and coefficient of determination ($R^2$) are adopted as the evaluation metrics of model performance. Lower values of MAE and RMSE indicate a better performance of the model. Conversely, the $R^2$, which varies between zero and one, evaluates the model's fit quality. A value of $R^2$ approaching one indicates a superior fit to the data.

**Baseline methods.** We conduct a comparative analysis of the proposed Ct-Transformer against two types of baseline methods: the incidence-based and Ct-based methods. The incidence-based method encompasses the EpiEstim [13], while the Ct-based method is further categorized into the statistical and deep learning methods based on their implementation strategies. In specific, the Ct-based statistical method includes the ViroSolver [25] and Regression [31]. As for the Ct-based deep learning method, we realize several prominent deep neural network architectures, namely the MLP [41], Transformer [45], and TFT [46] with the same input as the Ct-Transformer for comparison.

**Experimental environment.** Experiments are implemented in Python 3.10.12 with Pytorch 2.0.1. Each deep learning model is trained and tested on a workstation featuring an Intel(R) Core i5–13600KF CPU (@5.3 GHz), 32GB of RAM, and an NVIDIA RTX 4070Ti GPU with 12GB of memory. All deep learning models are trained over 30 epochs with the strategy of early stopping [62].

**Hyperparameter selection.** A grid search approach [63] is employed to determine the optimal hyperparameters for the deep learning models. The specific tuning spaces and the best values for each hyperparameter of the Ct-Transformer are presented in Table 2. Meanwhile, the explored tuning spaces and the best hyperparameter values for all deep learning models are provided in S5 Table in the S1 Supplementary Methods (4. Hyperparameters of Deep Learning Methods). The hyperparameters yielding the minimum validation loss for the Ct-Transformer are used in subsequent experiments to estimate $R_t$.

## Results

### Relationship between Ct values and $R_t$

As shown in Fig 2A, we find the ratio of the number of newly infected individuals to the currently infected individuals keeps high during the early stage of the epidemic. This ratio then

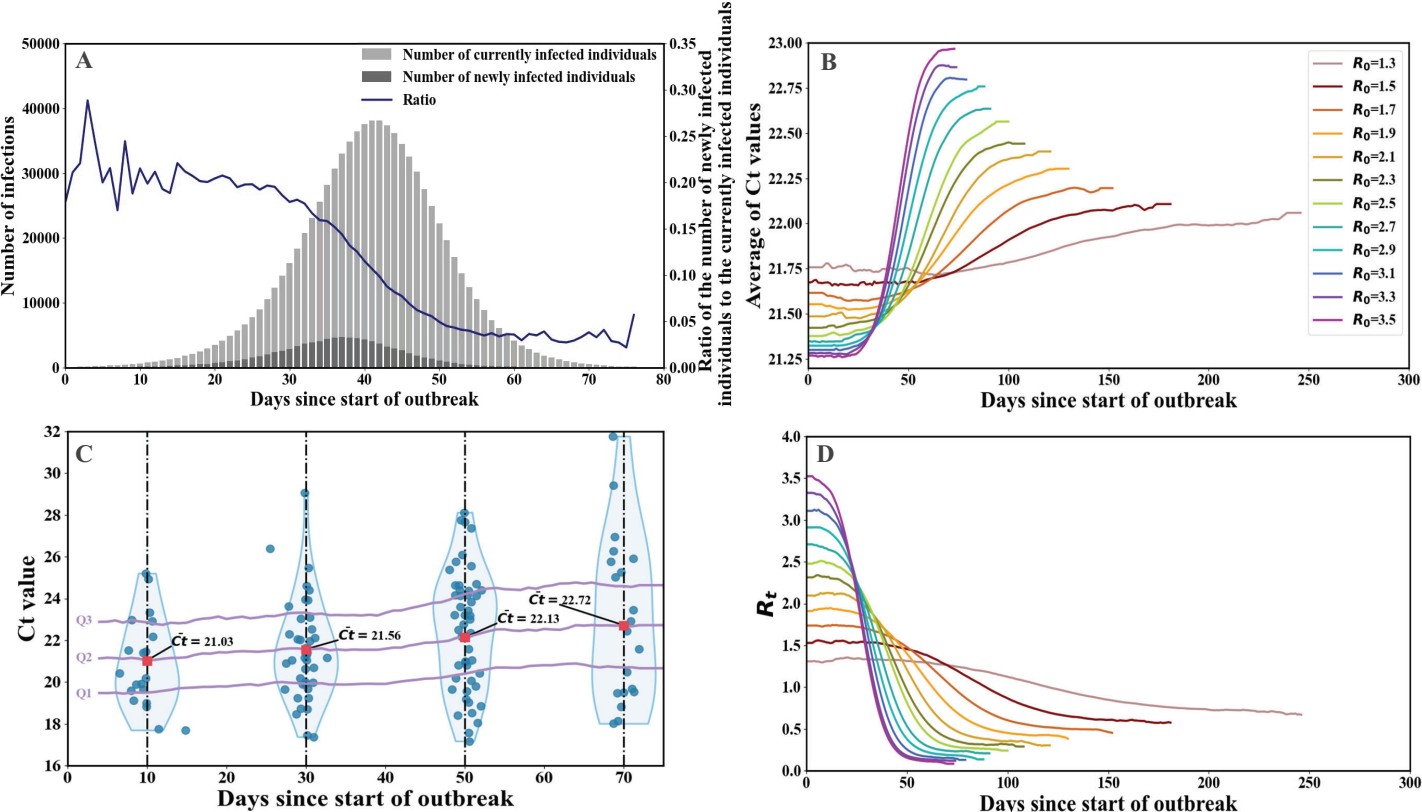

**Fig 2. The Ct values reflect epidemic dynamics throughout their outbreaks.** (A) The number of currently infected individuals (light histogram) and the number of newly infected individuals (dark histogram) in a stochastic simulation with $R_0$=3.0. (B) The average of Ct values for the infected population versus time. Each line shows the average of 1500 simulations. (C) The distribution of Ct values (violin plots) in the infected population, which are randomly selected on the detected days ($t$=10, 30, 50, and 70) during the outbreak described in (A). The median, along with the first and third quartiles of the distribution, are indicated by purple lines, while the red dots represent the average of Ct values (the median quartiles) on these detected days. (D) $R_t$ varies as the transmission of epidemics, with lines corresponding to those of the same color in panel (B). Each outbreak is simulated on the ER contact network.

decreases sharply as the epidemic reaches its peak, and finally keeps low towards the end of the epidemic. This trend implies that during the early stage of the epidemic, random sampling is more likely to detect individuals who have been recently infected and exhibit lower Ct values. Conversely, during the declining stage of the epidemic, the sampled infections present higher Ct values as they approach recovery. This is revealed in the average of Ct values in Fig 2C.

The temporal evolution of $R_t$ and the average of Ct values are respectively presented in Fig 2B and 2D. We find that $R_t$ keeps relatively stable during the early stage of the epidemic and then undergoes a rapid decline when reaching the peak. This trend is revealed by the evolution of the average of Ct values, which initially remains steady but then experiences a sharp increase as the transmission of the epidemic. In the meanwhile, a lower average of Ct values at the onset of the epidemic is found in the simulated epidemic with a higher $R_0$. These results imply the time-varying distribution of Ct values may serve as an observable proxy to infer the transmission dynamics of an epidemic [25].

## Supervised learning results

**Performance on the ER dataset.** We first evaluate the performance of the supervised Ct-Transformer on the ER dataset. The estimated $R_t$ for two stochastic simulations with different

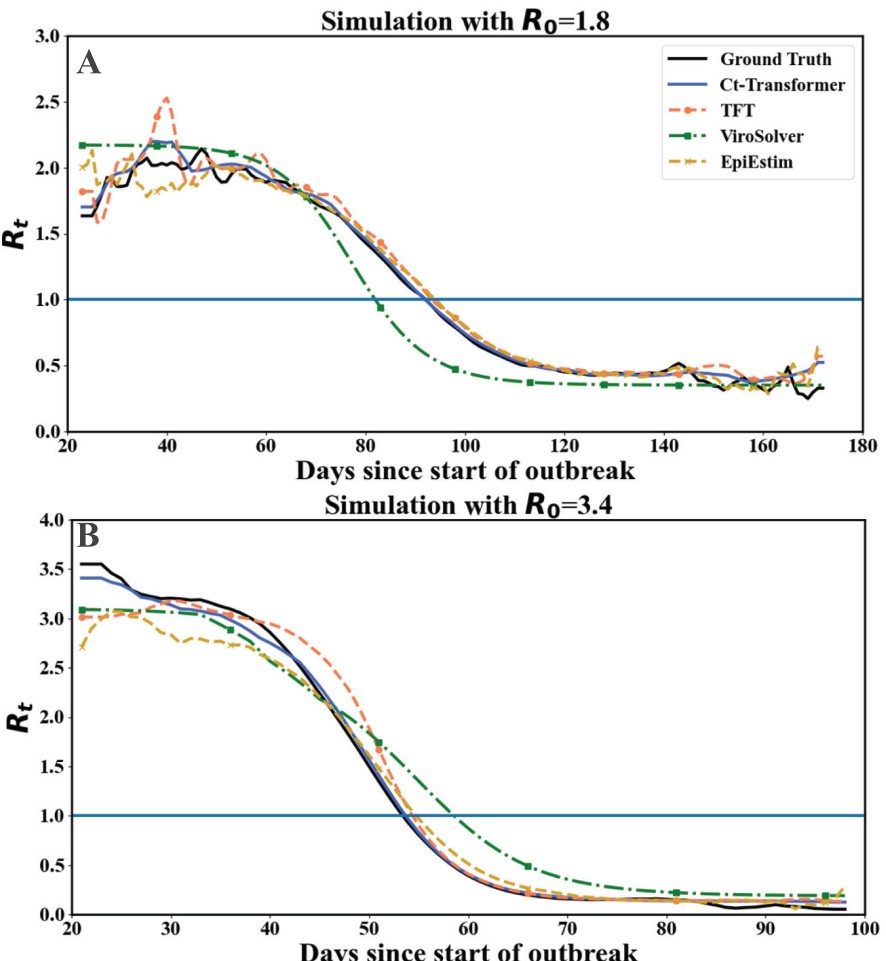

**Fig 3. The performance of the supervised Ct-Transformer on two stochastic simulations on the ER network.**
Panels (A) and (B) display the $R_t$ estimations by the Ct-Transformer and other alternative models for simulations
where $R_0$ is set to 1.8 and 3.4, respectively. To be noted that, all models begin estimating $R_t$ when there are fifteen cases
in the simulated population. The ground truth values are derived from the micro-transmission chain of the two
stochastic simulations, based on the definition of the case reproduction number. The estimation results of EpiEstim
(dotted yellow line), ViroSolver (dotted green line), and TFT (orange line) are presented as the representative
alternative methods for incidence-based, Ct-based statistical, and Ct-based deep learning approaches, respectively.

$R_0$ (i.e., $R_0$=1.8 and $R_0$=3.4) are presented in Fig 3. Overall, the supervised Ct-Transformer
(blue line) captures the temporal evolution of $R_t$ more accurately than both the incidence-
based and the Ct-based methods. The supervised Ct-Transformer shows enhanced precision
in estimating the infection point (i.e., the time when $R_t$ reaches one) than the ViroSolver and
TFT. In specific, the ViroSolver estimates the infection point significantly earlier than the
Ground Truth (black line) in the simulation with $R_0$=1.8. Conversely, it estimates the infection
point with a large delay in the simulation with $R_0$=3.4. Meanwhile, the performance of the
EpiEstim (dotted yellow line) exhibits bias at the beginning of both simulations. Further, the
95% confidence intervals of $R_t$ estimation for different stochastic simulations are shown in S2
Fig in S1 Supplementary Methods (5. Confidence Intervals of $R_t$ Estimation). We find that the
95% confidence intervals given by the Ct-Transformer can accurately capture the scales of the
temporal evolution of $R_t$.

**Table 3. Results of the supervised Ct-Transformer and baseline methods on the ER dataset.** We respectively show the average result of simulations with each $R_0 \in \{1.2,$ 1.8, 2.2, 2.8, 3.4\}$ in the testing set. The Average represents the average of these above results. For each $R_0$ and the Average, the best results are in bold and the second best are underlined.

| Test $R_0$ | | | $R_0=1.2$ | $R_0=1.8$ | $R_0=2.2$ | $R_0=2.8$ | $R_0=3.4$ | Average |
|---|---|---|---|---|---|---|---|---|
| Ct-based Deep Learning Method | Ct-Transformer | MAE | 0.083 | **0.060** | **0.058** | **0.044** | **0.063** | **0.062** |
| | | RMSE | 0.144 | **0.106** | **0.098** | **0.073** | **0.093** | **0.103** |
| | | $R^2$ | 0.801 | **0.974** | **0.987** | **0.995** | **0.995** | **0.950** |
| | TFT | MAE | 0.102 | 0.082 | 0.086 | 0.069 | 0.111 | 0.090 |
| | | RMSE | 0.177 | 0.110 | 0.138 | 0.108 | 0.173 | 0.141 |
| | | $R^2$ | 0.653 | 0.956 | 0.967 | 0.985 | 0.981 | 0.908 |
| | Transformer | MAE | 0.163 | 0.123 | 0.121 | 0.186 | 0.313 | 0.181 |
| | | RMSE | 0.245 | 0.190 | 0.182 | 0.253 | 0.479 | 0.270 |
| | | $R^2$ | 0.272 | 0.906 | 0.943 | 0.914 | 0.867 | 0.780 |
| | MLP | MAE | 0.180 | 0.096 | 0.130 | 0.189 | 0.281 | 0.175 |
| | | RMSE | 0.242 | 0.146 | 0.178 | 0.262 | 0.393 | 0.244 |
| | | $R^2$ | 0.440 | 0.953 | 0.958 | 0.946 | 0.914 | 0.842 |
| Ct-based Statistical Method | ViroSolver | MAE | 0.102 | 0.160 | 0.164 | 0.167 | 0.231 | 0.165 |
| | | RMSE | 0.158 | 0.206 | 0.238 | 0.209 | 0.274 | 0.217 |
| | | $R^2$ | 0.775 | 0.906 | 0.927 | 0.967 | 0.957 | 0.906 |
| | Regression | MAE | 0.200 | 0.119 | 0.140 | 0.186 | 0.170 | 0.163 |
| | | RMSE | 0.270 | 0.173 | 0.201 | 0.288 | 0.267 | 0.240 |
| | | $R^2$ | 0.282 | 0.928 | 0.944 | 0.903 | 0.951 | 0.802 |
| Incidence-based Method | EpiEstim | MAE | **0.067** | 0.081 | 0.092 | 0.128 | 0.155 | 0.105 |
| | | RMSE | **0.104** | 0.196 | 0.136 | 0.185 | 0.226 | 0.169 |
| | | $R^2$ | **0.905** | 0.924 | 0.977 | 0.975 | 0.971 | **0.950** |

The quantitative results of the model performance are summarized in Table 3. In terms of the **Average**, the supervised Ct-Transformer outperforms all the baseline methods in estimating $R_t$ despite a slight decline in performance when $R_0$ has the lowest value of 1.2. Compared with the best results (i.e., EpiEstim) achieved by both incidence-based and Ct-based statistical methods, the supervised Ct-Transformer attains a 41% decrease in MAE and a 39.1% reduction in RMSE. Further, in comparison with the optimal results (i.e., TFT) in other Ct-based deep learning methods, the supervised Ct-Transformer continues to exhibit superior accuracy, which achieves a 31.1% reduction in MAE, a 27.0% reduction in RMSE and 4.6% increase in $R^2$.

**Impact of detection rates on performance.** The testing policy associated with the availability of testing resources can be summarized as the detection rate [24], which changes during the course of the epidemic spreading. Meanwhile, the variations in detection rate lead to inaccurate time series data, which further causes the biases of the standard methods when estimating $R_t$, such as the EpiEstim [13]. To demonstrate the performance of the proposed Ct-Transformer, we implement five detection scenarios similar to [31], which are listed as follows:

1. **Full Detection:** a fixed detection rate of 100%. This scenario represents the ideal situation in which all infected individuals are detected;

2. **Scenario 1:** a fixed detection rate of 25%. This scenario represents the situation of stable detection;

3. **Scenario 2:** a fixed detection rate of 10%. This scenario represents the situation of stable but limited detection;

**Table 4. Results of the supervised Ct-Transformer and the EpiEstim on the ER dataset under different detection scenarios.** We show the average results of simulations with $R_0 \in \{1.2, 2.2, 3.4\}$ in the testing set. For each detection scenario, the best results are shown in bold between the Ct-Transformer and the EpiEstim.

| Test $R_0$ | | $R_0$=1.2 | | $R_0$=2.2 | | $R_0$=3.4 | |
|---|---|---|---|---|---|---|---|
| Dection | Method | Ct-Transformer | EpiEstim | Ct-Transformer | EpiEstim | Ct-Transformer | EpiEstim |
| Full Detection | MAE | 0.083 | **0.067** | **0.060** | 0.092 | **0.063** | 0.155 |
| | RMSE | 0.144 | **0.104** | **0.106** | 0.136 | **0.093** | 0.226 |
| | $R^2$ | 0.901 | **0.905** | **0.974** | 0.970 | **0.995** | 0.971 |
| Scenario 1 | MAE | **0.113** | 0.122 | **0.094** | 0.140 | **0.093** | 0.235 |
| | RMSE | **0.199** | 0.213 | **0.179** | 0.240 | **0.155** | 0.398 |
| | $R^2$ | **0.875** | 0.724 | **0.934** | 0.923 | **0.985** | 0.938 |
| Scenario 2 | MAE | **0.156** | 0.169 | **0.103** | 0.164 | **0.116** | 0.276 |
| | RMSE | **0.266** | 0.284 | **0.261** | 0.305 | **0.207** | 0.508 |
| | $R^2$ | **0.847** | 0.721 | **0.909** | 0.807 | **0.972** | 0.909 |
| Scenario 3 | MAE | **0.090** | 0.096 | **0.078** | 0.132 | **0.088** | 0.223 |
| | RMSE | **0.157** | 0.181 | **0.148** | 0.236 | **0.147** | 0.393 |
| | $R^2$ | **0.876** | 0.741 | **0.943** | 0.940 | **0.987** | 0.940 |
| Scenario 4 | MAE | **0.115** | 0.129 | **0.090** | 0.150 | **0.096** | 0.252 |
| | RMSE | **0.203** | 0.220 | **0.159** | 0.288 | **0.171** | 0.478 |
| | $R^2$ | **0.855** | 0.712 | **0.914** | 0.904 | **0.981** | 0.918 |

4. **Scenario 3:** a detection rate increases from 15% to 60% as the transmission of epidemics. This scenario represents the situation of expansion in detection;

5. **Scenario 4:** a fixed detection rate of 25% but 5% in the early stage of the epidemic (i.e., the early stage is defined as the first 20 days since the start of the outbreak). This scenario represents the situation of limited detection resources at the early stage of the epidemic.

We present the full comparisons between the Ct-Transformer and EpiEstim on the ER dataset under different detection scenarios, as shown in Table 4. We find the supervised Ct-Transformer consistently outperforms the EpiEstim across all detection scenarios, except the Full Detection scenario when $R_0$=1.2. In the meanwhile, compared to the Full Detection scenario, the increased percentage in MAE is minimal for the Ct-Transformer (see S3(A) Fig in the Scenario 1 to Scenario 4). These results are also robust when we compare the performance of Ct-Transformer and EpiEstim on the SF dataset (see S6 Table and S3(B) Fig in S1 Supplementary Methods (6. Further Exploration of Detection Rate on Performance)). These results demonstrate the proposed Ct-Transformer is robust to the time-varying detection rate and limited detection resources.

## Self-supervised learning results

In this section, we present the result of the self-supervised learning of Ct-Transformer on both the synthetic and real-world datasets. Notably, 30% of the patches are masked with zero at step (1) during training process of self-supervised learning, which can better learn representations from the distribution of Ct values. The performances of the Ct-Transformer with other rates of masked patches are reported in S7 Table in S1 Supplementary Methods (7. Rate of Masked Patches for Self-supervised Learning).

**Performance on the SF dataset.** We first obtain a pre-trained model by self-supervised training the Ct-Transformer on the ER dataset. Then, we implement both the Lin. Prob and End2End supervised training on part of SF dataset. The performance of the Ct-Transformer

**Table 5. Results of the Ct-Transformer (with End2End, Lin. Prob, and Sup.), other Ct-based supervised and incidence-based methods on the SF dataset.** We show the average results of simulations with $R_0 \in \{1.2, 1.8, 2.2, 2.8, 3.4\}$ in the testing set. The Average represents the average of the above results. For each $R_0$ and the Average, the best results are in bold and the second best are underlined.

| Test $R_0$ | | | $R_0$=1.2 | $R_0$=1.8 | $R_0$=2.2 | $R_0$=2.8 | $R_0$=3.4 | Average |
|---|---|---|---|---|---|---|---|---|
| Ct-Transformer | End2End | MAE | **0.094** | **0.094** | <u>0.081</u> | **0.082** | **0.074** | **0.085** |
| | | RMSE | <u>0.170</u> | **0.188** | **0.175** | **0.183** | **0.144** | **0.172** |
| | | $R^2$ | <u>0.958</u> | <u>0.973</u> | **0.984** | **0.986** | **0.994** | **0.979** |
| | Lin.Prob | MAE | 0.154 | 0.133 | 0.139 | 0.221 | 0.219 | 0.173 |
| | | RMSE | 0.233 | 0.246 | 0.293 | 0.470 | 0.387 | 0.326 |
| | | $R^2$ | 0.888 | 0.962 | 0.939 | 0.899 | 0.923 | 0.922 |
| | Sup. | MAE | <u>0.096</u> | <u>0.095</u> | **0.078** | <u>0.085</u> | 0.086 | 0.088 |
| | | RMSE | **0.167** | 0.195 | <u>0.177</u> | 0.198 | <u>0.164</u> | <u>0.180</u> |
| | | $R^2$ | **0.964** | 0.969 | <u>0.983</u> | <u>0.984</u> | <u>0.990</u> | <u>0.978</u> |
| Ct-based Deep Learning Method | TFT | MAE | 0.135 | 0.125 | 0.104 | 0.114 | 0.107 | 0.117 |
| | | RMSE | 0.247 | 0.233 | 0.223 | 0.246 | 0.197 | 0.229 |
| | | $R^2$ | 0.846 | 0.963 | 0.968 | 0.972 | 0.984 | 0.947 |
| | Transformer | MAE | 0.210 | 0.245 | 0.306 | 0.414 | 0.448 | 0.325 |
| | | RMSE | 0.333 | 0.404 | 0.554 | 0.786 | 0.816 | 0.579 |
| | | $R^2$ | 0.676 | 0.798 | 0.750 | 0.675 | 0.622 | 0.704 |
| | MLP | MAE | 0.271 | 0.211 | 0.168 | 0.308 | 0.378 | 0.267 |
| | | RMSE | 0.421 | 0.442 | 0.329 | 0.564 | 0.639 | 0.479 |
| | | $R^2$ | 0.754 | 0.866 | 0.945 | 0.900 | 0.891 | 0.871 |
| Ct-based Statistical Method | ViroSolver | MAE | 0.106 | 0.197 | 0.211 | 0.252 | 0.280 | 0.209 |
| | | RMSE | 0.171 | 0.288 | 0.377 | 0.418 | 0.441 | 0.339 |
| | | $R^2$ | 0.953 | 0.891 | 0.882 | 0.913 | 0.908 | 0.909 |
| | Regression | MAE | 0.161 | 0.218 | 0.263 | 0.324 | 0.372 | 0.268 |
| | | RMSE | 0.307 | 0.441 | 0.556 | 0.721 | 0.822 | 0.569 |
| | | $R^2$ | 0.866 | 0.843 | 0.828 | 0.815 | 0.805 | 0.831 |
| Incidence-based Method | EpiEstim | MAE | 0.120 | 0.122 | 0.121 | 0.171 | 0.208 | 0.148 |
| | | RMSE | 0.216 | <u>0.193</u> | 0.196 | 0.293 | 0.387 | 0.257 |
| | | $R^2$ | 0.932 | **0.978** | **0.984** | 0.970 | 0.958 | 0.964 |

(with Lin. Prob, End2End and the supervised training from scratch), other Ct-based supervised methods and the incidence-based methods on the SF dataset are summarized in Table 5.

In terms of the **Average**, the Lin. Prob performs better than the Ct-based statistical methods (i.e., ViroSolver and Regression) and two Ct-based deep learning methods (i.e., Transformer and MLP), with the reduction in MAE ranging from 17.2% to 46.8%. Meanwhile, the End2End performs slightly better than the Ct-Transformer trained from scratch and achieves the best result. Compared with the incidence-based method, the End2End exhibits a 42.6% decrease in MAE and a 33.1% reduction in RMSE. The performance of the Ct-Transformer, across all learning approaches, experiences a minor decrease in performance when $R_0$ is lower on the SF dataset, which is consistent with those observed on the ER dataset.

**Real-world data verification.** In this part, we present the result of the self-supervised Ct-Transformer with end-to-end fine-tuning on the real-world dataset. As shown in Fig 4A, the average of Ct values over time display a distinctly different trend compared to both the skewness of Ct values and the $R_t$ trend, as illustrated in Fig 4B. This behavior mirrors the phenomena observed in the synthetic dataset, as reported in Fig 2B and 2D. Specifically, the Ct-Transformer undergoes self-supervised training on the ER dataset. Then, this pre-trained

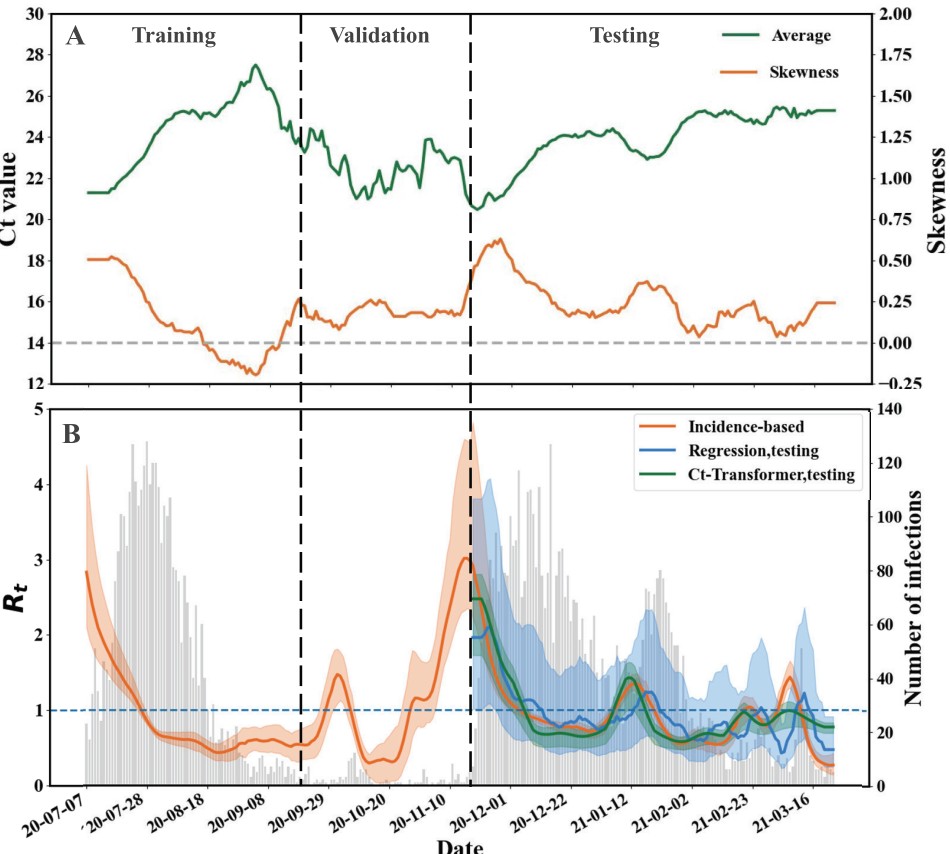

**Fig 4. Estimation results in the real-world dataset.** (A) Daily average (green line) and skewness (orange line) of Ct values from July 6th 2020 to March 23th 2021. The dotted horizontal line represents the value of skewness equal to zero. (B) The orange line and shaded area indicate the average and 95% confidence intervals for the results of the incidence-based method, while the blue line and shaded area indicate the average and 95% confidence intervals for the results of the regression method. The data mentioned above is provided by [31]. The green line and shaded area indicate the average and 95% confidence intervals for the results of the self-supervised Ct-Transformer. The grey bars represent the number of infections by sampling date. The overall period is divided into training, validation, and testing periods (left, middle, and right columns).

model is supervised fine-tuned with end-to-end approaches using the Hong Kong COVID-19 outbreak dataset. Distinct from its application on the synthetic datasets, the input for the Ct-Transformer in this real-world dataset is replaced with the average and skewness of Ct values.

As shown in Fig 4B, this real-world dataset is partitioned according to the method described in [31]. In the results of the testing set, the pre-trained Ct-Transformer with End2End fine-tuning can capture the temporal evolution of $R_t$ accurately, especially before Feb. 23rd 2021. In contrast and as reported in [31], the regression method, designed to estimate $R_t$ using the average and skewness of Ct values, attains a MAE of 0.25. In this context, the proposed self-supervised learning of Ct-Transformer achieves a smaller MAE of 0.19 and realizes a 24% reduction.

Leveraging the pre-trained Ct-Transformer by self-supervised learning on the synthetic datasets, we can selectively retrain the 'Prediction Head' or the entire model over fewer epochs to accurately estimate $R_t$ across diverse datasets, including the SF dataset and the Hong Kong COVID-19 dataset. This approach enhances the model's generalization capability and concurrently diminishes both computational demands and data requirements.

## Ablation experiments

To quantify the benefit of each layer in the architecture of Ct-Transformer, we perform the following ablation experiments and quantify the percentage increase in loss compared to the original architecture:

1. **No patch:** We set the length of patch in the Patching layer to one.

2. **No CVS:** We replace the CVS layer with a linear layer, which performs a linear transformation on the input.

3. **No TFE:** We directly remove the TFE layer.

4. **No GRN:** We directly remove the GRN layer.

5. **No MSA:** We set the head-shared weights for the $V$ matrix in Eq (5) to be head-specific, which is calculated as:

$$\tilde{H} = \frac{1}{h_m}\sum_{h=1}^{h_m} A(QW_Q^{(h)}, KW_K^{(h)})VW_V^{(h)} \tag{9}$$

We train the ablated model using the hyper-parameter settings as the same in Data. The average percentage increase in loss compared to the original architecture of Ct-Transformer is calculated as follows:

$$\Delta \bar{E}^{\alpha} = \frac{\sum_{r \in R} \frac{E_r^{\alpha} - E_r}{E_r}}{n_R} \times 100\% \tag{10}$$

where $\Delta \bar{E}^{\alpha}$ represents the average percentage increase in loss (MAE or RMSE or $R^2$) of the ablated model for type $\alpha$ (No patch or No CFS or No TFE or No GRN or No MSA). $E_r^{\alpha}$ and $E_r$ respectively represent the average loss of the ablated model for type $\alpha$ and the original architecture of Ct-Transformer in the simulated epidemics with $R_0$ equal to $r$, where $r \in R = \{1.2, 1.8, 2.2, 2.8, 3.4\}$. $n_R$ represents the number of $R_0$ in the set $R$. The average percentage increase in loss (MAE, RMSE, and $R^2$) for each type of ablation is shown in Fig 5.

**Patching.** As shown in Fig 5A, removing the Patching results in a 15.9% increase in MAE when averaged by the results on both the ER and SF datasets. In addition to improving the computational efficiency, the Patching layer crucially emphasizes the differences among all time steps, which aids in extracting the temporal features. Further analysis of this phenomenon, together with the exploration of the patch lengths, is detailed in S1 Supplementary Methods (8. Analysis of the Patching Layer).

**Temporal Features Extraction.** From Fig 5, the most important role of the TFE layer in the architecture of the Ct-Transformer becomes evident. In specific, the removal of the TFE layer results in a significant increase in MAE, RMSE and $R^2$, which are respectively as 54.7%, 49.8% and -3.0%. These results highlight the crucial role of extracting temporal features from the time-varying distribution of Ct values for the accurate estimates of $R_t$.

**Nonlinear fitting and dependency learning.** The GRN and the MSA layers have a large impact on the performance of Ct-Transformer on the SF dataset. In specific, the removal of the GRN and MSA layers respectively results in a 20.7% increase and 30.2% increase in MAE on the SF dataset, while these become 4.8% and 2.7% in the ER dataset, as shown in Fig 5A. These differences reflect the importance of non-linear fitting and dependency learning in heterogeneous contact patterns.

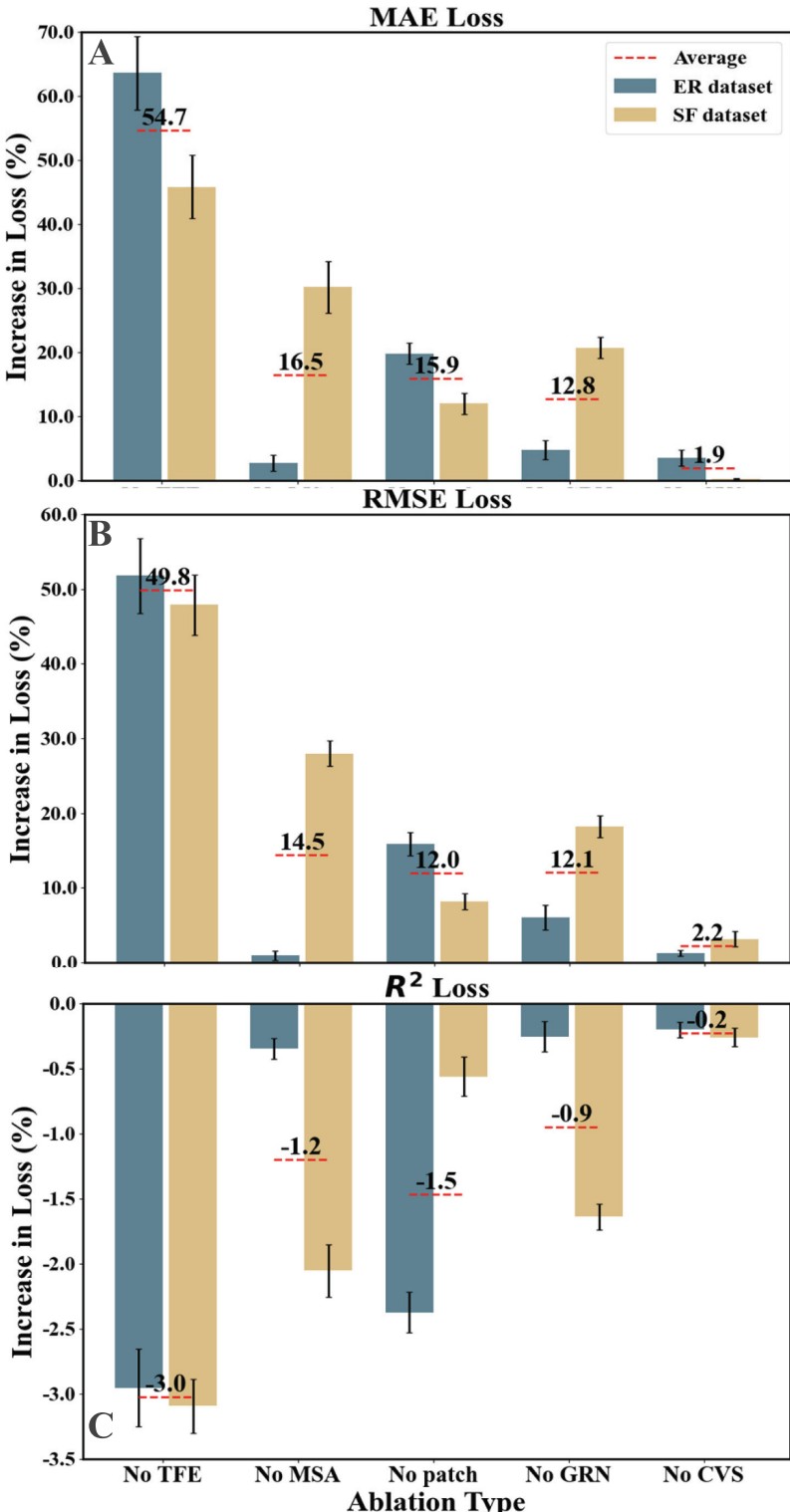

**Fig 5. The average percentage increase in loss for each ablation type $\Delta \bar{E}^x$.** (A), (B), and (C) respectively represent the MAE, RMSE, and $R^2$ loss. The gray and yellow bars respectively represent the ablation results on the ER and SF datasets. The numbers on the red dotted line represent the average result of both datasets. The error bars represent the standard deviation among the increases in loss for each $R_0$ in the testing set.

In summary, each layer in the Ct-Transformer improves the performance of estimating $R_t$. The removal of the TFE layer results in the most significant percentage increase in loss, while removing the CVS layer leads to the least percentage increase in loss. The effects of removing the GRN and MSA layers differ across the two synthetic datasets.

## Discussion

Real-time estimates of $R_t$ are crucial for understanding epidemic dynamics and timely adjustment of interventions to mitigate and prevent the spread of emergent infectious diseases. In this work, we propose the Ct-Transformer, an attention-based deep neural network architecture for estimating the time-varying effective reproduction number $R_t$ based on the time-varying distribution of Ct values. We find the Ct-Transformer outperforms both the incidence-based and existing Ct-based methods in estimating $R_t$. This work highlights the effectiveness of the Ct-based deep learning method in improving real-time estimates of $R_t$ and timely tracking of epidemic dynamics.

Some studies [25, 29–33] have developed the statistical methods to estimate $R_t$ based on the distribution of Ct values of the infected population. The ViroSolver [25] employs Bayesian inference to estimate $R_t$ by collecting the distribution of Ct values at single or multiple time points in the population. This method leverages the idea of refining the prior knowledge with new evidence and re-estimating the probability distributions of the results continuously, which significantly enhances interpretability. However, this method needs the prior knowledges including both the transmission dynamic such as the prior distribution for the epidemic seed time, the constant exponential growth of infection incidence, etc. In contrast, the proposed deep learning model, Ct-Transformer, can estimate $R_t$ accurately without these assumptions. In specific, the supervised Ct-Transformer is trained using the simulated outbreaks with four $R_0$, while it can estimate $R_t$ accurately in the simulated outbreaks with a range of $R_0$. This capability is attributed to the layers designed in the architecture of the Ct-Transformer, which can extract the temporal features from the time-varying distribution of Ct values and establish the complex non-linear relationship with $R_t$ in a purely data-driven manner. Further, we explore and compare the impact of different combinations of Ct variables on the model performance to fully leverage the information from the evolution of Ct values, as detailed in S1 Supplementary Methods (3. Ct Variables in the Synthetic Datasets). The distribution of Ct values with an interval of 4 can better reflect the shape of the distribution and the average can serve as the viral load level of the infected population. Thus, the combination of these two variables enables the Ct-Transformer to more effectively capture the evolution of Ct values compared to the ViroSolver and regression model [31], which rely on a single type of Ct variables input.

To demonstrate the robustness of the proposed Ct-Transformer, we have evaluated the performance of the supervised Ct-Transformer under different detection scenarios. We find the proposed model is capable of accurately estimating $R_t$ under both the low and time-varying detection rates. In contrast, the incidence-based methods, such as the EpiEstim [13], exhibit fluctuations and instability during the early stages of epidemics. These sensitive analyses present that the proposed Ct-Transformer is robust to detection resources, which can be explained by the input of the Ct-Transformer and the advantages offered by deep learning methods. Such robustness is especially valuable in the epidemic area with limited detection resources or during the early stage of the new emergent epidemic.

To further enhance the adaptability of the proposed model, we have developed the self-supervised learning of Ct-Transformer, which can be pre-trained on the synthetic dataset without labels and then undergo supervised training on a small target dataset. Our findings on

the synthetic dataset indicate that the self-supervised Ct-Transformer through End2End fine-tuning strategy achieves a comparable performance with the model supervised trained from scratch. Further, this approach yields superior performance on the Hong Kong COVID-19 dataset, evidenced by lower MAE loss and narrower confidence intervals compared to the regression model proposed in [31]. These results prove that the developed pre-trained Ct-Transformer can be adopted to estimate $R_t$ across diverse epidemic dynamics through fine-tuning strategies.

All layers in the architecture of Ct-Transformer contribute to the accuracy of estimating $R_t$, as presented by the result of ablation experiments. The Ct-Transformer outperforms both the Transformer [45] and MLP [41], which is primarily attributed to the crucial role of extracting the temporal features through a GRU network in the TFE layer. Additionally, this model benefits from an innovative patching mechanism and exhibits superior performance compared with the TFT [46]. The Ct-Transformer's ability to accurately estimate $R_t$ in stochastic simulations with heterogeneous contact networks is further enhanced by the incorporation of the gated unit in the GRN layer and self-attention mechanisms in the MSA layer.

There are several limitations in our work. Firstly, the Ct-Transformer needs prior knowledge about the evolution of Ct values in the newly emergent epidemic, which may be uncertain despite the reduction of extensive training datasets due to self-supervised learning. Additionally, the trajectories of Ct values can vary significantly across different populations. In particular, populations with higher vaccination rates [64] or with younger age [65] exhibit higher average Ct values during the epidemic outbreak. This intrinsic relationship between Ct values and $R_t$ requires recalibration when applying the Ct-Transformer to a new outbreak or the same infectious disease in different countries with distinct contact patterns. Regarding the architecture of the model, there is potential for further improving the Ct-Transformer by developing an adaptive patching strategy based on the input, which may enhance performance beyond the current fixed length of the patch. Lastly, there are two kinds of definition on time-varying effective reproduction number: the instantaneous reproduction number and the case reproduction number. The former focuses on the short time windows and very sensitive to immediate changes in the transmission, while the latter reflects more on the actual transmission rather than the potential. Although the proposed Ct-Transformer performs well in estimating the case reproduction number, whether it will be effective in estimating the instantaneous reproduction number left for further investigations.

## Conclusion

In this paper, we propose a novel attention-based deep neural network architecture Ct-Transformer and the self-supervised learning approach for the estimates of the time-vary effective reproduction number $R_t$ based on the time-varying distribution of Ct values. The supervised learning of Ct-Transformer outperforms standard incidence-based, Ct-based statistic approaches and other Ct-based deep learning methods. These results indicate the effectiveness of the designed architecture of Ct-Transformer in extracting the temporal features and processing the nonlinear relationships. The pre-trained Ct-Transformer is evaluated on both the synthetic and real-world datasets and attains a comparable performance with the supervised model. We demonstrate the Ct-based deep learning model can improve the real-time estimates of $R_t$, especially in monitoring the newly emerged infectious disease. Further, the architecture of Ct-Transformer can indeed provide insights for designing models in other tasks, including time series forecasting.

## Supporting information

**S1 Supplementary Methods. 1. Parameters of the Agent-based SEIR Transmission Model and Ct Value Model.** 2. Calculation of $R_t$ based on Micro-transmission Chains. 3. Ct Variables in the Synthetic Datasets. 4. Hyper-parameters of Deep Learning Methods. 5. Confidence Intervals of $R_t$ Estimation. 6. Further Exploration of Detection Rate on Model Performance. 7. Rate of Masked Patches for Self-supervised Learning. 8. Analysis of the Patching Layer.
(ZIP)

**S1 Fig. The trajectories of Ct values.** The gray lines represent the trajectories of Ct values for 50 randomly selected infected individuals, while the black line represents the trajectory of one infected individual.
(EPS)

**S2 Fig. The 95% confidence intervals of $R_t$ estimation.** The pink line and shaded area respectively represent the average of $R_t$ and the 95% confidence intervals.
(EPS)

**S3 Fig. The increased percentage in MAE relative to the Full Detection scenario in stochastic simulations on both ER and SF networks with $R_0$=2.2.** The greyish-blue and yellow bars respectively represent the MAE loss of the Ct-Transformer and EpiEstim in different detection scenarios. The numbers displayed on the bars indicate the increased percentage in MAE relative to the Full Detection scenario.
(EPS)

**S4 Fig. Attention maps of the supervised Ct-Transformer with or without patching.** (A) Attention map in stochastic simulation on the ER network with patching. (B) Attention map in stochastic simulation on the SF network with patching. (C) Attention map in stochastic simulation on the ER network without patching. (D) Attention map in stochastic simulation on the SF network without patching. All stochastic simulations with $R_0$=1.2.
(EPS)

**S5 Fig. The MAE loss with varying patch lengths $P$ = [2, 4, 6, 8, 10, 12, 14, 16, 18] on the ER and SF datasets.**
(EPS)

**S1 Table. Parameters of the agent-based SEIR transmission model and Ct value model.**
(PDF)

**S2 Table. Intervals $d$ and the set $D$ in the distribution of Ct values.**
(PDF)

**S3 Table. The sensitivity results of intervals $d$ on ER and SF datasets.** andThe **Average** means the average of simulations with $R_0 \in$ {1.2, 1.8, 2.2, 2.8, 3.4} in the testing set. For each $R_0$ and the **Average**, the best results are in **bold** the runners-up are presented as <u>underlined</u>.
(PDF)

**S4 Table. The sensitivity results of Ct variables (Var) on ER and SF datasets.** The **Average** means the average of the simulations with $R_0 \in$ {1.2, 1.8, 2.2, 2.8, 3.4} in the testing set. For each $R_0$ and the **Average**, the best results are in **bold** and the runners-up are presented as <u>underlined</u>.
(PDF)

**S5 Table. Hyperparameters, tuning spaces, and the best hyperparameter settings for deep learning methods (Ct-Transformer, TFT, Transformer, and MLP) on the ER dataset and**

SF dataset.
(PDF)

**S6 Table. Results of the supervised Ct-Transformer and the EpiEstim method on the SF dataset under different detection scenarios.** For each detection scenario, the better one is presented as in **bold**.
(PDF)

**S7 Table. The sensitivity results of mask rates on the SF dataset.** The **Average** means the average of the simulations with $R_0 \in \{1.2, 1.8, 2.2, 2.8, 3.4\}$. For each $R_0$ and the **Average**, the best one is in **bold** and the runners-up is presented as underlined.
(PDF)

## Author Contributions

**Conceptualization:** Gui-Quan Sun, Quan-Hui Liu.

**Data curation:** Xin-Yu Zhang, Lan-Lan Yu, Wei-Yi Wang.

**Formal analysis:** Xin-Yu Zhang, Gui-Quan Sun, Jian-Cheng Lv, Tao Zhou, Quan-Hui Liu.

**Investigation:** Xin-Yu Zhang, Gui-Quan Sun, Jian-Cheng Lv, Tao Zhou, Quan-Hui Liu.

**Methodology:** Xin-Yu Zhang, Lan-Lan Yu.

**Visualization:** Xin-Yu Zhang, Wei-Yi Wang.

**Writing – original draft:** Xin-Yu Zhang, Quan-Hui Liu.

**Writing – review & editing:** Xin-Yu Zhang, Lan-Lan Yu, Wei-Yi Wang, Gui-Quan Sun, Jian-Cheng Lv, Tao Zhou, Quan-Hui Liu.

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
