## [Decision Letter · Decision Letter 0]

27 Jul 2024

Dear Liu,

Thank you very much for submitting your manuscript "Estimating the time-varying effective reproduction number via Cycle Threshold-based Transformer" for consideration at PLOS Computational Biology.

As with all papers reviewed by the journal, your manuscript was reviewed by members of the editorial board and by several independent reviewers. In light of the reviews (below this email), we would like to invite the resubmission of a significantly-revised version that takes into account the reviewers' comments.

I endorse the recommendations by both reviewers who suggest the following points for improvements: a more comprehensive discussion of the methodology and experimental setup and a better explanation on the scope of contributions; clarification on parameters, math notations, and abbreviations; citation of related existing work on effective reproduction number estimation.

We cannot make any decision about publication until we have seen the revised manuscript and your response to the reviewers' comments. Your revised manuscript is also likely to be sent to reviewers for further evaluation.

Sincerely,

Claudio José Struchiner, M.D., Sc.D.

Academic Editor

PLOS Computational Biology

Thomas Leitner

Section Editor

PLOS Computational Biology

I endorse the recommendations by both reviewers who suggest the following points for improvements: a more comprehensive discussion of the methodology and experimental setup and a better explanation on the scope of contributions; clarification on parameters, math notations, and abbreviations; citation of related existing work on effective reproduction number estimation.

Reviewer's Responses to Questions

**Comments to the Authors:**

Reviewer #1: The review is uploaded as an attachment.

Reviewer #2: The review is uploaded as an attachment

**Have the authors made all data and (if applicable) computational code underlying the findings in their manuscript fully available?**

Reviewer #1: **No: **The authors provide all data and code in a GitHub repository that is not open to public yet.

Reviewer #2: **No: **The authors have mentioned that the data/codes will be available on Github upon acceptance of the manuscript.

PLOS authors have the option to publish the peer review history of their article (what does this mean?). If published, this will include your full peer review and any attached files.

Reviewer #1: No

Reviewer #2: No
---

## [Decision Letter · Decision Letter 1]

4 Dec 2024

Dear Liu,

We are pleased to inform you that your manuscript 'Estimating the time-varying effective reproduction number via Cycle Threshold-based Transformer' has been provisionally accepted for publication in PLOS Computational Biology.

Best regards,

Claudio José Struchiner, M.D., Sc.D.

Academic Editor

PLOS Computational Biology

Thomas Leitner

Section Editor

PLOS Computational Biology

Feilim Mac Gabhann

Editor-in-Chief

PLOS Computational Biology

Jason Papin

Editor-in-Chief

PLOS Computational Biology

Reviewer's Responses to Questions

**Comments to the Authors:**

Reviewer #1: The authors have clearly and thoroughly solved all my confusions including the concepts, experimental settings, and suggestions on relevant literatures and abbreviation definitions. There are a few places where the authors mention "fine-turning", which might be "fine-tuning". Please fix it if it is a typo, or ignore it if it is not.

Reviewer #2: The authors have addressed all the major and minor comments satisfactorily. I believe that this manuscript has improved. Happy for this to be accepted for publication.

**Have the authors made all data and (if applicable) computational code underlying the findings in their manuscript fully available?**

Reviewer #1: Yes

Reviewer #2: Yes

PLOS authors have the option to publish the peer review history of their article (what does this mean?). If published, this will include your full peer review and any attached files.

Reviewer #1: No

Reviewer #2: No

---

## [Editor Report · Acceptance letter]

11 Dec 2024

PCOMPBIOL-D-24-00718R1 

Estimating the time-varying effective reproduction number via Cycle Threshold-based Transformer

Dear Dr Liu,

I am pleased to inform you that your manuscript has been formally accepted for publication in PLOS Computational Biology. Your manuscript is now with our production department and you will be notified of the publication date in due course.

With kind regards,

Zsofia Freund
